# Post Audit of Groundwater Model Predictions under Changing Conditions

Jacob Kidmose * , Lars Troldborg and Jens Christian Refsgaard

Geological Survey of Denmark and Greenland, Øster Voldgade 10, 1350 Copenhagen, Denmark
* Correspondence: jbki@geus.dk; Tel.: +45-91-33-36-14

**Abstract:** Post audits of hydrological or groundwater models are the last part of the modelling protocol, where the original model predictions are tested using new data obtained after a certain period. The evaluation of model predictions and associated predictive uncertainty was performed by comparing an original hydrological model, a model with post audited geology, and a model with post audited geology and calibrated against new types of observation data. The post audit showed original model predictions close to what was observed (in terms of abstracted volumes necessary to lower a shallow groundwater table). In contrast to the robust original model predictions, the original model underestimated the predictive uncertainty compared to the assessments of uncertainty using the new and updated post audit model. To ensure a robust model evaluation, we propose a four-step post audit protocol, including (1) testing the validity of the original model predictions with new data, (2) estimating the predictive uncertainty of the original model, (3) producing a new post audit model(s) based on revising the conceptual model and calibration, and (4) assessing the predictive uncertainty of the new post audit models. The work presented here was motivated by the lack of studies that, after a certain time, have re-evaluated model predictions (post audit) with new data.

**Keywords:** post audit; protocol; groundwater model; hydrological model; predictive uncertainty; model uncertainty; groundwater





## 1. Introduction

A hydrological model is often applied to support water management decisions, by making predictions of future conditions, e.g., impacts of changes in water abstraction, infrastructure, land use, or climate. Tests of a hydrological model, to document whether it can simulate conditions similar to its intended applications, are crucial for establishing model credibility [1,2]. Such tests may be denoted validation [3], verification [4], evaluation [5], or history matching [6]. Despite the differences in terminology, it is commonly agreed that such tests (henceforth denoted model validation) can never result in a model being universally valid [3,6]; rather the ability of a site-specific model to perform certain predictions (model validity) is conditional on specific locations, acceptable accuracy/uncertainty levels, and the type of application [5,7].

Although thoroughly conducted model validation tests can significantly enhance a model's credibility with respect to making specific predictions within a certain accuracy, they are no guarantee that the model predictions will match the future realities. The ultimate test is therefore to wait some years and collect new field data for evaluating the extent to which the original model predictions match reality. Such evaluation is denoted post-audit validation [4]. Post audits are seldom performed [8]. For groundwater-related predictions, less than a dozen post audits have been published in the international literature. The reported post audits dealt with groundwater flow problems [9–13], as well as contaminant or reactive transport [8,14–17]. The general lessons learned from the post audits are that model predictions often comprise larger errors than expected, and that these errors often

are related to conceptual hydrogeological model errors and inaccurate estimates of future stresses, such as the recharge, abstraction, and contaminant loading rates.

Post audits are very valuable in several respects. First, revisiting the predictions after some years, when new field data are available, and evaluating the accuracy of the original model predictions is useful to contribute to the general understanding of the potentials and limitations of hydrological model predictions. Although each modelling study is unique, and generalizations therefore cannot be made from a single study, well-documented post audits represent the only true validation tests of model predictions. Hence, experiences from a large number of post audits would be useful for providing information about the situations where models typically fail. Anderson and Woessner [4], who reviewed five post audits, provided the only review paper published so far. Second, a post audit can help to improve our understanding of both the model and the system that is being modelled. This is particularly valuable in cases where a model is applied to make long-term predictions and the post audit is performed at an intermediate point in time [4]. In such cases, the new field data can be used to conduct a new round of model calibration, model predictions, and uncertainty assessments, which, due to the additional data and improved understanding, can be expected to result in reduced prediction uncertainties. Thus, Karlsen et al. [17] concluded that their post audit and recalibration, 25 years after the original predictions, led to an improved model conceptualization and enhanced predictive performance.

Model validation tests must be designed to test a model's ability to perform the kind of tasks for which it is specifically intended [3]. If the model is intended to make predictions for conditions that are assumed not to change from the calibration period to the prediction period, a simple and very straightforward split-sample (SS) test is adequate [18]. If, on the other hand, the model is intended to simulate situations that are beyond those prevailing in the calibration period, e.g., predictions of the impacts of changes in climate, land use, or water abstraction, then a so-called differential split-sample (DSS) test is required [18]. DSS tests imply calibrating on one situation and validating on another situation with changed conditions. As data are usually not available for the changed situations for which predictions are required, such DSS tests need to be carried out in other (e.g., neighboring) areas, where data from before/after the changes exist. This inevitably weakens the test, because the conditions in the neighboring areas are not identical to the area in question. Therefore, post audits are particularly relevant in cases where DSS tests are required.

Most of the post audits reported in the literature are SS type of tests, where no changes are assumed in climate, land use, or groundwater abstraction, and where long time series often are available for calibration [10,13,14,16,17]. In the two other studies [9,11], small changes (<50%) occurred in groundwater pumping, and these tests may be characterized as DSS-light. Andersen and Lu [12] is the only reported post audit dealing with significant changes in the groundwater abstraction scheme, and as such is the only strong DSS test that has been subject to a post audit.

Modelling practices have evolved significantly since the first post audits of electric analogue models were published three decades ago [9,10]. When Anderson and Woessner [4] introduced post audits as part of a modelling protocol, the state of the art involved trial-and-error calibration procedures and uncertainty assessments confined to sensitivity analyses. Today, uncertainty assessment is a fundamental element of good modelling practice [2,19,20]. The study by Karlsen et al. [17] is the only post audit employing the present state-of-the-art techniques, such as inverse modelling (using new field data) and assessment of prediction uncertainties.

The objectives of the present study were (i) to provide an example of a post audit for prediction of situations representing large extrapolations from the calibration situation (strong DSS test); (ii) to analyze how a post audit can improve the accuracy and reliability of model predictions; and (iii) to propose a protocol for performing post audits.

The post audit presented here involved a model used to design a motorway construction buried in an aquifer. During the subsequent construction work, excavations allowed a post audit of the geological interpretation, while monitoring of pumping rates

and drawdowns provided data for a post audit of the original groundwater model. In addition to such a traditional post audit study, assessments were made of how the prediction uncertainties were affected by the new post audit data. While using standard techniques for model calibration and uncertainty analyses, the novelty of the study lies in the post audit study and the proposed new protocol.

## 2. Materials and Methods

### 2.1. Post Audit—Project Context

This post audit study was made possible through close collaboration with the Danish Road Agency, who during the nine years 2010–2019 designed, constructed, and post-monitored a new motorway through the city of Silkeborg in Jutland, Denmark. To reduce noise pollution to neighbors and to avoid cutting local roads, the motorway was lowered below ground surface over a 2 km stretch. In this way, the motorway was lowered into an aquifer with a shallow groundwater table. The motorway was designed to last at least to the end of the century (~yr 2100). As previous studies in the region had indicated that, by year 2100, climate change might impact groundwater tables through increases of more than one meter in many areas [21], the Road Agency decided to assess the climate change impacts on groundwater conditions as a basis for the motorway design. This was the basis for the original groundwater model study reported by Kidmose et al. [22]. During the nine-year project period, other versions of the original model were also developed. This included a version coupled with MOUSE, to simulate the interaction between urban runoff networks and the upper groundwater table [23]. Furthermore, a version of the original model was used to test the data assimilation of groundwater hydraulic heads and stream discharge with an ensemble Kalman filter algorithm, to reduce the model bias in a simulated real-cast modelling framework [24].

As the groundwater conditions were critical for the motorway design, and as climate change conditions represent a significant extrapolation beyond existing climate conditions, it was relevant for the Road Agency to assess the robustness of the conclusions from the original study using a post audit study, where the stress conditions differ significantly from today's conditions. This was possible because the construction works implied considerable excavations, exposing some of the geology and requiring major abstractions and groundwater table drawdowns. Monitoring during the construction period formed a unique dataset that was exploited in the subsequent post audit study reported here.

### 2.2. Study Site and Data

The study site is located in the central part of the mainland of Denmark, where hydrological conditions are humid, North-Western European, with an average precipitation of 903 mm yr-1, temperature of 7.3 °C, and reference evapotranspiration of 545 mm yr-1 (1961–1990, DMI; [25]). The geology consists of glacial clay-tills, outwash sand deposits in the upper 10–30 m, and Miocene marine silts and clays with only a few embedded sandy lenses below the glacial layers. The hydrogeological unit in focus was the outwash sand, called terrace sand, deposited in the Gudenå river valley system, Figure 1. The terrace sand is the upper unconfined aquifer. The Gudenå river formed the southern model boundary for the local hydrological models used in the study.

A comprehensive amount of high-quality data were available for the study. The aquifer had previously been used for water supply (now abandoned due to groundwater pollution), so data from several old boreholes and pumping test existed. In addition, a large number of geotechnical boreholes and four stream discharge stations were established in connection with the motorway project. Altogether, single groundwater head values from 97 wells and daily values for more than two years (2010–2012) of groundwater heads from 35 wells, as well as discharge from the four gauging stations, were collected and used in the original hydrological model [22].

For the post audit study, continuous time series data from most of the 35 wells and four gauging stations were available (a few of the wells had been dug away during the

construction work). The groundwater table is generally only 2–5 m below the surface. As part of the construction work, sheet piling was installed 10–15 m below the surface and the bottom of the motorway construction was down to $1\frac{1}{2}$ meters below the groundwater table. To allow concrete casting to take place in dry conditions, drawdown of the groundwater table was necessary during part of the construction period. This was done by pumping water from drains in the construction pit. Data on the amount of water pumped away were collected from the contractor performing the road construction.

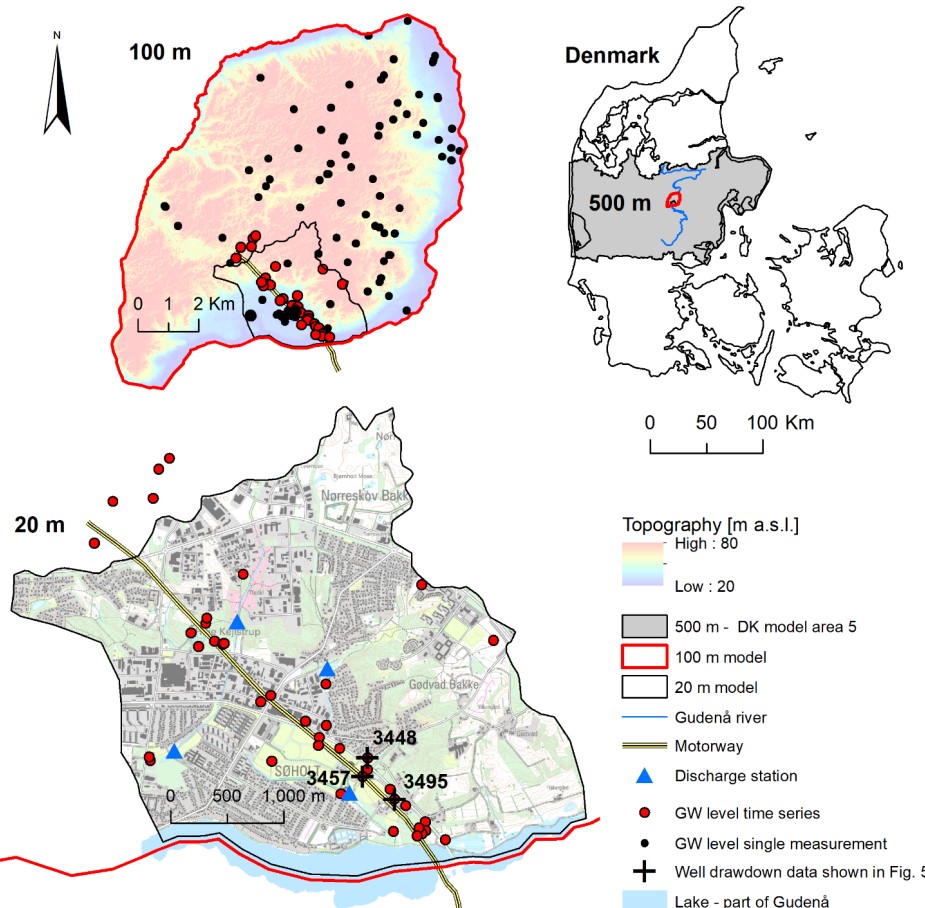

**Figure 1.** Location of the study area and nested hydrological models. Danish National Water Resources model, area 5 with 500 m grid (**top right**). The 100 m hydrological model (**top left**) and local hydrological model in the city of Silkeborg, with 20 m grid.

### 2.3. Hydrological Model Configurations

The hydrological model code MIKE SHE [26,27] was selected, because of its ability to simulate transient groundwater flow with a fully coupled simulation of evapotranspiration and unsaturated zone flow in daily time steps, relevant for the prediction of annual groundwater fluctuation. MIKE SHE was coupled to MIKE 11, allowing transient simulations of flows in streams, creeks, and rivers. With simulations of daily changes in groundwater recharge, groundwater levels, and stream flow over several years, the models also represented seasonal changes. In the original model, 3 years of dynamic (non-steady state) simulations from 2010 to 2012 were created.

Three hydrological model configurations were analyzed. They differed in terms of the availability of geological information and the data used for model calibration, but had the same hydrological setup (boundary conditions, surface water system, forcing with time series of daily precipitation, temperature, and reference evapotranspiration).

- Baseline model. This was the original hydrological model developed to support the design of the motorway [22].
- Post audit-Geology model. Here, the geological model as a part of the hydrological model was updated based on new geological insights inferred from observing the exposed slopes in the excavations. This model was calibrated using the same data as available for the Baseline model.
- Post audit-Hydrogeology model. This hydrological model used the geology from the Post audit-Geology model. In addition, the groundwater pumping and drawdown data from the construction period were included in the dataset used for model calibration.

### 2.4. Geology

The geological model used in the Baseline model was developed for the area of the 100 m model, Figure 1. This was based on existing geological models, primarily the Danish National Water Resources Model [28,29], geophysical data, borehole data from the Danish borehole database "Jupiter", and geotechnical borehole data along the motorway. The 3D geological model was used in the 20 m models and the 100 m hydrological models, Figure 1. Important hydrogeological features are illustrated in Figure 2, the terrace sand, its vertical and horizontal extent, the hydraulic contact between the upper glacial meltwater sand and the terrace sand, and finally the groundwater flow between the terrace sand and the underlying Miocene layers.

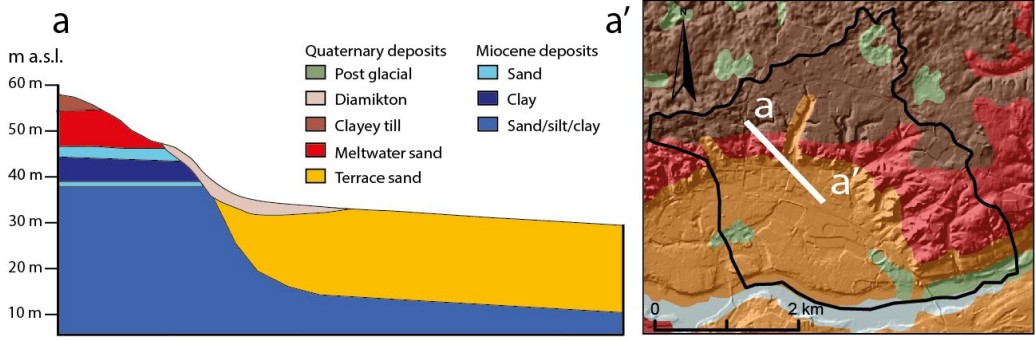

**Figure 2.** Conceptual geological model based on Jakobsen et al. [30]. The right side shows the location of the profile and the surface geology. The Miocene succession, illustrated with blue colors, consists of (from the top): sand (Addit member), and below from the Vejle Fjord Formation, clay, sand, and fine sand/silt/clay layer. The geological profile a-a' shown to the left are marked as a white line a-a' on the map to the right.

The first step in the post audit was to update and revise the geological model. This was done using observations from two field campaigns; one in the spring of 2014 and one in the fall of 2014, where the excavation of the incised motorway transect exposed important geological features. Investigations were documented in 14 composite sedimentological logs, together with an overview photogrammetry of the sites [30]. After including the updated geology, the hydrological model was recalibrated using the same calibration data as in the Baseline model. This became the Post audit-Geology model.

### 2.5. Hydrological Models

The Baseline, the Post audit-Geology, and the Post audit-Hydrogeology (hydrological) models all follow the same nested modelling approach with a regional model having a 500 × 500 m grid discretization (500 m model), a 100 m model inheriting boundary conditions from the 500 m model, and a local 20 m model covering the northern part of Silkeborg City and inheriting boundary conditions from the 100 m model, Figure 1. The modelling approach was the same as applied in [22], but the results from the 20 m model were not used in this previous study because the finer 20 m discretization was not necessary to assess general climate change impacts on groundwater levels. In the present

study, it was necessary to use the 20 m grid discretization, because the area between the sheet piling, where groundwater levels were lowered during construction, had a width of close to 20 m. With a 20 m model, the area of the groundwater lowering is more precisely simulated. The 500 m model with an area of 12,500 km$^2$ is one of seven sub-models of the National Danish Water Resources hydrological model that cover Denmark (43.000 km$^2$). The model is bounded by the sea towards the East and West (specified head = 0) and by hydrological catchment boundaries toward the North and South (no flows). Calibration was performed for the period 2000–2003 for 2592 groundwater head observation wells and 66 discharge time series, both as part of the Danish national monitoring program (NOVANA). The three hydrological models all run in daily timesteps, with daily forcing of precipitation in 10 × 10 km gridded input, temperature in 20 × 20 km gridded input, and references evapotranspiration in 20 × 20 km gridded input. Details of the hydro-stratigraphy, calibration, and hydrological model setup of the National Danish Water Resources model can be found in Henriksen et al. [28] and Højberg et al. [13]. The 100 and 20 m models cover 103 km$^2$ and 15 km$^2$, respectively. Both models have the same vertical grid-discretization, with three saturated zone numerical layers; the lower one consists of pre-quaternary layers, the middle numerical layer, the glacial meltwater sand, the most upper numerical layer, the terrace sand in the river valley, and clayey till above 45 m a.s.l. The diamikton is also a part of the upper numerical layer, see Figures 1 and 2. The 100 m model has an open specified head boundary condition for the lower numerical layers, with daily changing hydraulic heads from the 500 m model for the entire period the models are run. Similarly, the 20 m model inherits boundary conditions from the 100 m model, but for all three numerical layers. Regarding the numerical grid discretization, several tests were done to test the consistency of the model predictions independently of the model grid. Among these, a 50 m grid version was tested and reported by Kidmose et al. [23]. Furthermore, a 10 m version was tested instead of the used 20 m version. The 10 m version proved (at the time of setting up the original baseline model) to be too difficult to calibrate because of the computational burden. The 100 and 20 m models have an open general head boundary condition toward the lake at the southern boundary of the models. Observed daily water stages of the lake were used for this general head boundary condition.

### 2.6. Model Calibrations

Model calibrations were performed with the automatic optimizer PEST [31]. Calibration data used for the Baseline model and the Post audit-Geology comprised single measurements of the hydraulic head (*n* = 97) and time series from 2010 to 2012 of hydraulic heads (*n* = 35) and discharge (*n* = 4). For the Post audit-Hydrogeology time series data were available for the period 2010 to 2016; and in addition, data were available for water quantities pumped from the construction pits. During model calibration, the 100 m and 20 m models were run successively, and parameters were updated in both models for each parameter test. Simulated groundwater head observations for optimization were extracted from the 100 m model, with drawdown data and water volumes from the 20 m model.

The objective function was defined as the root mean squared error of nine weighted observation groups, Table 1. Most weights were placed on the annual level and fluctuation of the hydraulic head. Twelve parameters were selected for calibration based on a sensitivity analysis. The result of the sensitivity analysis for the different models was almost identical, but because of the different model geology and objective functions (described below), there were some differences. To perform the best inter-model comparison, it was decided to calibrate using the same parameters. In the calibration of the Post audit-Hydrogeology model, observation data from the drawdown curves and abstracted groundwater volume data from the construction period were utilized. Thereby, two new groups were added to the objective function. PumpMax describes the error in simulated pumped water volumes in 2014 and PumpGrad the gradient of the drawdown curves in 2014 (Table 1). The estimated parameter values for all three models are shown in Appendix A and commented on in the results section below.

**Table 1.** The objective function and nine embedded groups used for the three models.

| Objective function=$\sum_i(w_i\times HTS\_ME)^2+\sum_j(w_j\times Hobs\_mean)^2+\sum_k(w_k\times HTS_{ErrAmpl})^2+\sum_l(w_l\times Qbal_{Winter})^2+$ $\sum_m(w_m\times Qbal_{Spring})^2+\sum_n(w_n\times Qbal_{Summer})^2+\sum_s(w_s\times Qbal_{Autumn})^2+(w_p\times AbsTotal)^2+\sum_g(w_g\times PumpGrad)^2$ | | | | | |
|---|---|---|---|---|---|
| **Group** | **Definition** | **Initial Group Weight** | | | **No. Obs.** |
| | | **Baseline Model** | **Post Audit-Geology Model** | **Post Audit-Hydrogeology Model** | |
| HTS_ME | Mean error of time series of hydraulic head (daily) | 1534 | 984 | 984 | 35 |
| Hobs_mean | Error of average h for the period 1990–2010 * | 525 | 498 | 498 | 97 |
| HTS_ErrAmpl | Error of maximum annual amplitude of h (daily) | 50 | 50 | 50 | 30 |
| Qbal_Winter | Mean seasonal error of discharge (Dec. Jan. Feb.) | 20 | 20 | 20 | 4 |
| Qbal_Spring | Mean seasonal error of discharge (Mar. Apr. May) | 5 | 5 | 5 | 4 |
| Qbal_Summer | Mean seasonal error of discharge (Jun. Jul. Aug.) | 20 | 20 | 20 | 4 |
| Qbal_Autumn | Mean seasonal error of discharge (Sep. Oct. Nov.) | 154 | 101 | 101 | 4 |
| PumpMax | Error of abstracted water amounts | 0 | 0 | 751 | 1 |
| PumpGrad | Error in gradient on drawdown curves | 0 | 0 | 751 | 6 |

Note:* Single measurements of hydraulic head compared with simulated average h for the period.

The initial group weight, Table 1, is the total impact (phi) on the objective function from the nine individual groups. Before calibration, the residuals (observed—simulated) were higher for the baseline model than with the updated geology in the post audit geology and post audit hydrogeology models. The initial group weight or impact from time series and single measurements would then be higher for the baseline model, even though $w_i$ and $w_j$ were the same for all model optimizations. Similarly, autumn discharge was also affected by the changes in geology between the models.

*2.7. Post Audit of the Geological Conceptualisation*

Prediction uncertainties were assessed with parameter uncertainty using the PEST tools RANDPAR and PNULPAR in a Monte Carlo analysis of predictive uncertainty [31]. The PEST tools were applied to generate 100 model parameter sets based on optimized model parameters, a lognormal distribution of the parameter uncertainty, 95% confidence limits, and a logical parameter order relationship. This repetition of assessing the predictive uncertainty for both the pre- and post-audit models facilitated a general assessment of how uncertainty is conceived in models before and after a model post audit process.

**3. Results**

*3.1. Post Audit of the Geological Conceptualisation*

The geological post audit focused on describing the terrace sand, Figure 3C, and its possible hydraulic connection to the meltwater sand above the several-meters-thick clay layer from the Vejle Fjord formation, Figure 3A. In the original geological interpretation of the Vejle Fjord clay layer, it was undulating both in vertical location and thickness. The outcrops illustrated a very consistent thickness and vertical location of the layer. Based on the exposed areas of the hillslope, Figure 3A,B, the hydraulic connection between the terrace sand and the upper sand layers (Meltwater sand and upper Miocene sand) was estimated to be poor. This estimation was also supported by the hydraulic head observations from these sand layers, supporting a groundwater table 5 to 8 m higher than for the terrace sand. The outcrops, especially the Miocene strata, were furthermore investigated by leading Danish Miocene experts and chrono stratigraphically dated using micro-fossils as biomarkers. Further documentation of the geological investigation and interpretation can be found in Jakobsen et al. [30]. The update of the geological model, based on the exposed geological features during excavation, resulted in a confirmation of the existing knowledge with minor corrections, in terms of lithological and hydro-geological descriptions. Knowledge of the geological chrono stratigraphic frame and formation environment of the pre-quaternary sediments increased significantly from the geological post audit. Although this update did not affect the hydro-geological interpretation directly, the geological model should be

considered more mature, in terms of its credibility and trustworthiness, as well as in terms of applicability for a hydrological model.

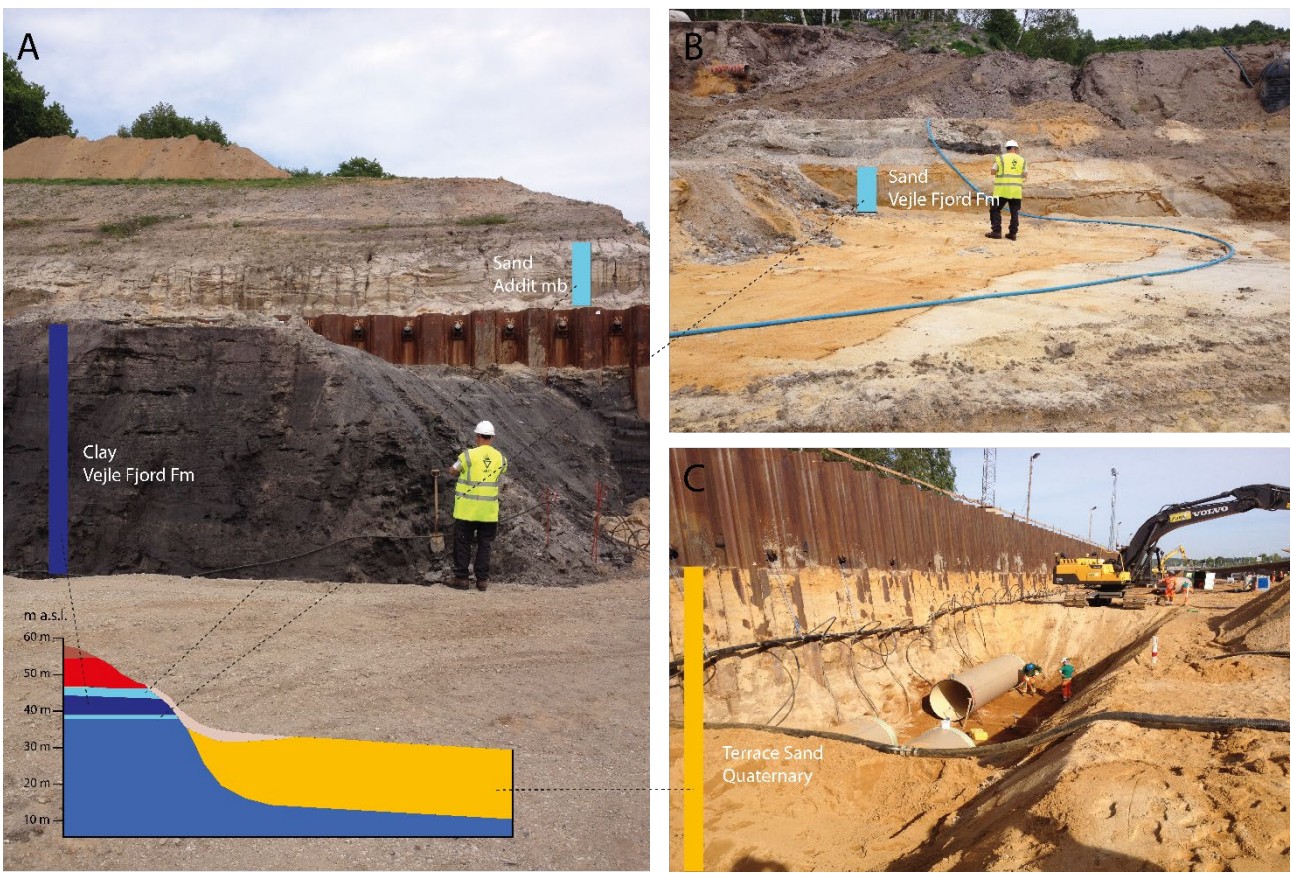

**Figure 3.** Post auditing the geological model on site. Sheet piling is visible in (**A**,**C**). Miocene sand from the Vejle Fjord Fm (**B**) During construction, the groundwater table was lowered in several sections along the motorway with a suction cell system at the sheet piling, with suction cells every 1 m connected to larger pumps, picture (**C**).

An overview of the geological outcrops exposed during the construction of the motorway and analyzed in the geological post audit is illustrated in Figure 3.

### 3.2. Hydrological Model Calibrations

The results of the parameter optimization are shown in three tables in the Appendix A. As expected, because the conceptual (geological) models and the data available for calibration differed between the three model calibrations, the estimated parameter values also varied. The confidence intervals were in all cases overlapping and, with a couple of exceptions, the estimated parameter values for each of the three models fell within the confidence intervals of the other two models. The hydraulic conductivity of the terrace sand, Figure 3C was analyzed with drawdown tests, with values of $1.15 \times 10^{-3}$ m/s. The calibrated parameter values of the horizontal hydraulic conductivity of the terrace sand were $1.06 \times 10^{-3}$ (Baseline model), $9.90 \times 10^{-4}$ (Post audit-Geology) and $1.64 \times 10^{-3}$ m/s (Post audit-Hydrogeology), with narrow 95 % confidence limits for all calibrations between $6.25 \times 10^{-4}$ and $2.39 \times 10^{-3}$ m/s. The precise fit of the model optimization is not surprising, considering 24 of 35 hydraulic head time series were screened in this aquifer.

The performances of the three models are illustrated in Table 2, with respect to the metrics used to define the objective function shown in Table 1. The statistical differences between the three models revealed that the inclusion of the objective function of drawdown curves and the amount of abstracted water in the Post audit-Hydrogeology model reduced

the model precision for the mean error (ME) and root mean squared error (RMSE) of the simulated hydraulic head. This is a trade-off, because the Post audit-Hydrogeology model gives less weight to the ME and RMSE metrics to optimize the PumpMax and PumpGrad objective function groups.

**Table 2.** Model performance.

| Metric | Baseline | Post Audit-Geology | Post Audit-Hydrogeology |
|---|---|---|---|
| ME of time series of hydraulic head | 1.03 m | 0.57 m | 1.07 m |
| RMSE of time series of hydraulic head | 2.38 m | 1.79 m | 2.26 m |
| ME of average h for the period 1990–2010 | 1.13 m | 1.29 m | 0.67 m |
| Error of maximum annual amplitude of h | 0.159 m | 0.150 m | 0.145 m |
| Mean seasonal error of discharge (December January February) | 12.2 L/s | 13.4 L/s | 12.3 L/s |
| Mean seasonal error of discharge (March April May) | −23.5 L/s | −19.3 L/s | −46.7 L/s |
| Mean seasonal error of discharge (June July August) | 42.1 L/s | 48.7 L/s | 26.5 L/s |
| Mean seasonal error of discharge (September October November) | 28.5 L/s | 29.5 L/s | 22.6 L/s |
| Error of abstracted water amounts | - | - | 0.640 mm |
| Mean error of drawdown curves | - | - | 0.055 m |

The errors of the amounts of abstracted water and drawdown were only calculated for the Post audit-Hydrogeology model, where these were included in the objective function. Abstracted water quantities are in units of millimeters (volume of abstracted water volume divided by the size of the model domain), and the error in drawdown curves in meters was calculated as the difference in the hydraulic head during the drawdown period.

The error of amount of abstracted water was reduced to practically zero in the Post audit-Hydrogeology optimization, with an observed value of 1237.59 mm and a simulated value of 1236.95 mm. This was possible with only one observation in this group and the prediction being directly controlled by the hydraulic conductivity and specific yield of the terrace sand parameters included in the calibration.

Figure 4 illustrates the spatial variation of absolute errors for the hydraulic heads from the time series (HTS_ME), together with a scatterplot of the hydraulic head single measurement observations (Hobs_mean).

The most prominent difference between the model calibrations was seen between the Baseline model and the Post audit-Geology model for the hydraulic head time series. The Post audit-Geology model showed a significant improvement of both ME and RMSE, where the ME was almost halved and RMS also reduced by 50 cm for the hydraulic head time series. Even though the geological update should have improved the simulated hydraulic head, the scale of the improvement was surprising, because the original geological model was thoroughly developed. The single-head measurements, with less weight in the objective function, were not improved between the Baseline and Post audit-Geology models, but were slightly decreased. As explained and shown in Figure 3, the geological update was performed close to the motorway where the time series wells are located and not in over entire 100 m model domain represented by the single head measurements. Including the drawdown data for model calibration in the Post audit-Hydrogeology model increased the ME and RMS compared to the Post audit-Geology model. It is well known in multi-objective parameter optimization that adding new groups in the objective function will result in a trade-off [32]. In this case, the optimization of the objective function in the Post audit-Hydrogeology model resulted in an improvement with respect to the two new objective function groups, while the old groups, e.g., the mean error on hydraulic head time series, yearly fluctuation of hydraulic head, and single hydraulic head observations had relatively less weight and hence a reduced accuracy. Compared to the Baseline model, the Post audit-Hydrogeology model showed a similar performance, with a 4 cm higher ME and 12 cm lower RMSE.

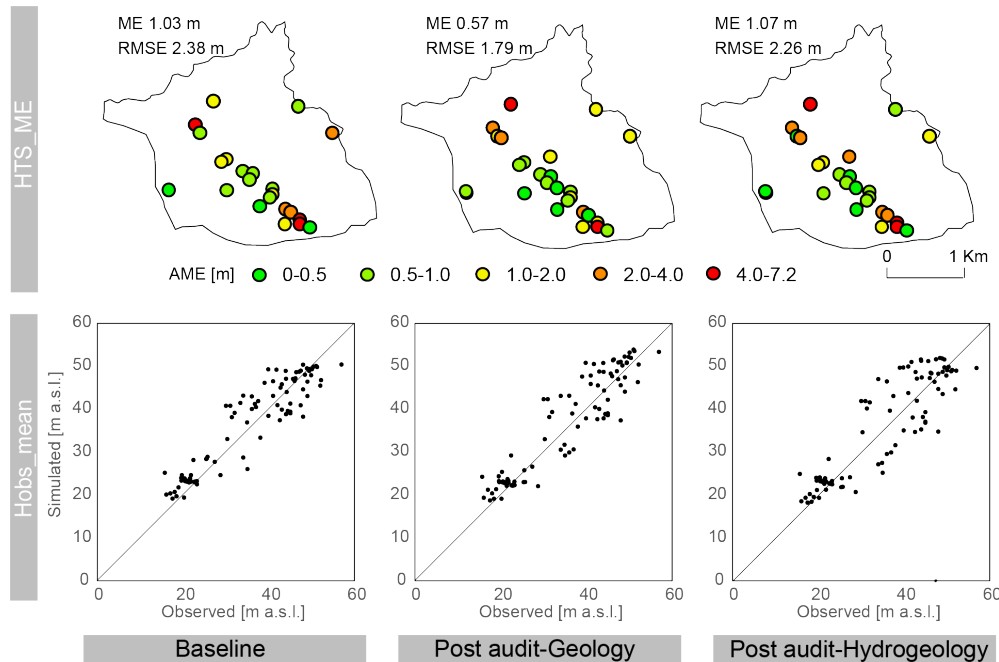

**Figure 4.** Spatial distribution of model precision of hydraulic heads for wells with observed time series for the Baseline model (**left**), the Post audit-Geology model (**middle**), and the Post audit-Hydrogeology model (**right**). The spatial representation (**upper maps**) is based on the HTS_ME group and shows the absolute mean error (AME), whereas the statistics are the mean error (ME) and root mean squared error (RMSE). Errors for the Hobs_mean group are illustrated in the lower scatter plots.

*3.3. Post Audit Tests of Hydrological Models*

The post audit tests were chosen to assess the ability of the models to simulate the water volumes abstracted from the construction pits around the motorway transect and the drawdowns in three wells (Figure 1) close to the abstraction sites. Most of the abstraction took place over $5\frac{1}{2}$ months (10 July 2014–22 December 2014) with variations over time and along the motorway transect. Daily data on abstraction rates distributed spatially along the transect do not exist. The abstraction was monitored by the contractor as accumulated volumes over weeks, with total quantities for all suction cells accumulated, Figure 3C. Even though the total period of active suction cells was noted, the spatial distribution of abstraction within a week was not. In the model, the conditions were simulated by keeping the hydraulic heads fixed at the levels specified for the suction cells, in the construction process, and abstraction volumes were thus simulated and could be compared with the monitored volumes. Due to a lack of detailed data, the groundwater heads were lowered to constant levels during the $5\frac{1}{2}$ months, without considering variations along the transect, or with time, caused by the development of the construction work.

Figure 5 shows simulated drawdown curves for the three wells, together with the predictive uncertainties caused by the parameter uncertainty. It is noted that the observations for well 3448 and well 3495 did not capture the maximum drawdowns, because the wells ran dry. It is seen that the simulated drawdown started at the same time for all three wells, while the observed drawdown started one month earlier for well 3495 and a couple of weeks earlier for well 3457 compared to well 3448. This reflects the fact that the construction work (and hence the abstractions and drawdowns) started at the south-eastern side and moved towards the north-west. This spatial difference was not available in the data used to force the model.

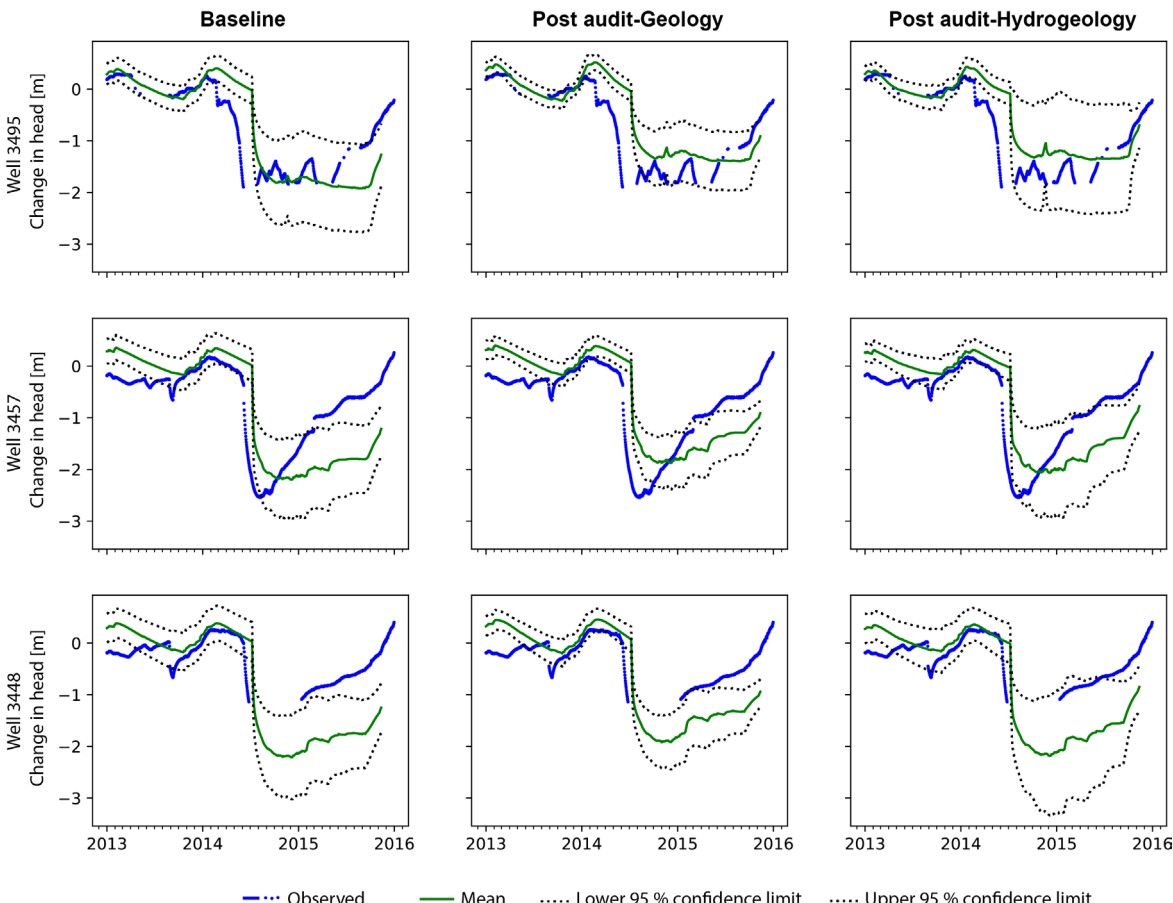

**Figure 5.** Simulated and observed drawdown curves at three wells for the three different models. Mean value, 95% upper, and lower confidence limits for the ensemble. Observed head data were not recorded because the well water levels were below the pressure transducer in well 3495 ($<-2.0$) and well 3448 ($<-1.2$).

Considering the above data limitations, the slopes and the maximum level of the drawdown curves were simulated quite well. The rebound curves, on the other hand, were not reproduced to the same extent. The complexity of the real drawdown was likely simplified when simulated, and therefore fine temporal pumping stops and starts were not simulated precisely, e.g., observed vs. simulated hydraulic head August 2014 to November 2015 at well 3495.

Figure 6 shows the pumped groundwater volumes used to lower groundwater to specified levels during the construction of the motorway. The predicted volumes (mean of 100 Monte Carlo simulations) were within 20% of the observed abstracted volumes for all three models. For construction works such as this, it is considered satisfactory if model predicted volumes can keep within a factor 2 of the subsequent real-work abstractions. Hence, the results shown in Figure 6 are remarkably good.

With drawdowns of $2-2\frac{1}{2}$ meters compared to annual head variations typically of 15 cm, this represents a very strong post audit test. With a quite good simulation of drawdowns and abstraction volumes, the results from the post audit can, overall, be considered quite good.

The predictive uncertainty, illustrated as standard deviations of the model ensemble and confidence limits, varied between the models. The Post audit-Hydrogeology model clearly showed the largest uncertainty, whereas the Post audit-Geology model had the most narrow range of predictions. The predictive uncertainty indicated by the spread of the ensemble for the three models simulating the volume of pumped groundwater also showed a wider range for the Post audit-Hydrogeology model, Figure 6.

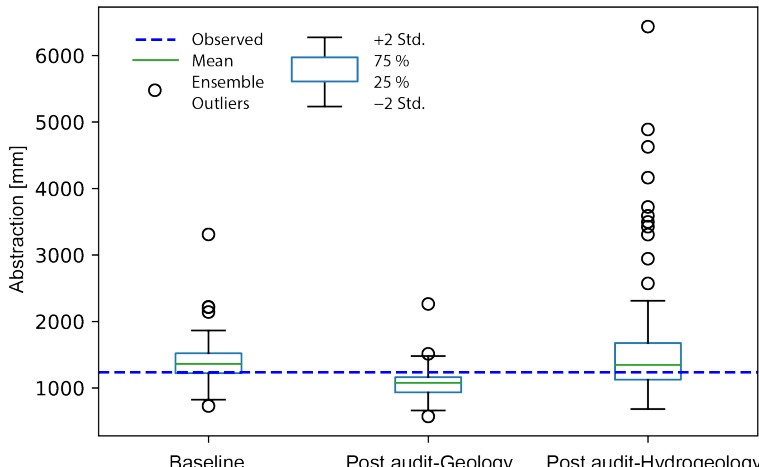

**Figure 6.** Simulated abstracted groundwater for the three models. The box plot shows the mean, quantiles, and two standard deviations, while the dots indicate the Monte Carlo ensemble members outside two standard deviations. The blue and dotted line shows the observed abstraction.

As well as the higher predictive uncertainty of the post audit-hydrogeology model, the model also had more outliers in terms of abstracted water, Figure 6. The horizontal hydraulic conductivity for the terrace sand was obviously a dominant parameter for the prediction of the drawdowns in this aquifer. For this parameter, more Monte Carlo realizations of the Post audit Hydrogeology model produces high values, thus resulting in more outliers with high abstraction. This, furthermore, originated from the higher upper confidence limit of the parameter from calibration of the Post audit-Hydrogeology model, compared to the other models. The same relation between high outliers and the hydraulic conductivity of the terrace sand was seen between the baseline model and the post audit geological model.

## 4. Discussion and Conclusions

The post audit described herein illustrates a modelling procedure with several steps, where the model setup and parameter optimization concept were altered over a long period, 2010–2019. The considerable resources and long timespan available in our study are uncommon. The unique possibilities of revisiting original model predictions and associated uncertainty and confronting them with new data are likewise uncommon. This provided a unique opportunity to elaborate, not only on how well the original model performed but also to what extent the uncertainty predictions based on the original model and data were in agreement with the uncertainty assessments made in a post audit context based on better data and knowledge.

### 4.1. This Study—Post Audit for Differential Split-Sample (DSS) Test

The model results indicated that the Baseline model predictions were quite close to the predictions from the Post audit-Geology and Post audit-Hydrogeology models. There was not a large evolution in the mean estimates between the models. This is in contrast to results from the other DSS tests reported in the literature [33], typically showing considerable errors. The reason that our predictions were relatively good could be the relatively large effort put into the original model, regarding (1) setup of the geological model, based on previous geological models of the area, geophysics, borehole data from Jupiter (the Danish national borehole archive), and new geotechnical boreholes that are primarily along the motorway transect; (2) the knowledge of aquifer characteristics (e.g., from pump tests); (3) the large number of hydraulic head times series with frequent data sampling (at least one time per day) and time series of stream discharge; and (4) sufficient time to develop a hydrological model with high confidence. It is further believed that the choice of a dynamic integrated groundwater–surface water model has an advantage over a steady, non-dynamic

model, where dynamic parameters such as specific yield and storage cannot be quantified. In the early calibration formulations, the objective functions were formulated without calibrating against the annual range of the hydraulic head time series (e.g., annual max. hydraulic head minus annual min. hydraulic head). Adding the annual range of hydraulic heads significantly improved the model simulation, which would not have been possible with a steady-state groundwater model.

For the application of the model to construct a motorway below the groundwater table, the assessment of model uncertainty is important, because of the planned long-term lifetime of the motorway (until year 2100) and the need to account for climate change. It is relevant to investigate uncertainty related to the prediction of drawdown and abstracted water amounts, because applying a hydrological model for predictions of future conditions represents a large extrapolation from the model conditions during calibration and validation. It will rarely be possible to test the future conditions, and therefore testing and calibrating the model to other extrapolations such as drawdown and water volumes is reasonable. The post audit gave confidence in the model representation of the aquifer properties and parameters; however, it did not give confidence regarding the model's ability to simulate far future groundwater recharge. The uncertainty analyses showed that the understanding of parameter uncertainty is strongly determined by the paradigms of calibration, e.g., simplicity of the objective function. It was originally expected that the predictive uncertainty interval resulting from the Post audit-Hydrogeological model, applying all the new post audit data, would be more narrow than the predictive uncertainty of the Baseline model. This turned out not to be the case; instead, the predictive uncertainty interval increased, even when the calibration statistics (RMSE and ME) were improved because of the geological update in the phase 2 model. Uncertainty predictions based on parameter uncertainty alone, as in this case, are conditioned on the conceptual model being correct, which will never be a correct assumption. In this case, the post audit data provided additional information about the groundwater system, which revealed some inconsistencies in the assumption of a perfect conceptual model and hence resulted in an increased uncertainty. This observation supports the findings of many other studies suggesting that uncertainty predictions, considering only parameter uncertainty, often underestimate uncertainties in predictions of situations that are extrapolations on the basis of calibration [20].

*4.2. Comparison with Previous Post Audits*

Some of the first post audits of groundwater models resulted in a rejection of the original model predictions. Konikow [9] showed, in an early post audit study, that the original model predictions were incorrect, because of incorrect assumptions about pumpage, and most likely also because of model structural error, such as a two-dimensional solution to a three-dimensional groundwater flow problem (a conceptual error). Anderson and Woessner [4] reviewed the five available post audits in the early 1990s and argued that the main reasons for the failure of the original model predictions was errors in the conceptual models or in defining the future stresses (e.g., assumptions of pumpage as in the Konikow study, [9]). Stewart and Langevin [11] found low predictability in original model predictions from 1981, mainly because of wrong assumptions about the hydrological systems response-time to reach steady conditions after stress was put on the system, e.g., a conceptual error. Recently, Brkic et al. [13] performed a post audit study confirming original model predictions from 1993 of groundwater recharge and pumping. A similar outcome for post audit results, confirming original predictions, was illustrated in the Silkeborg study. A possible reason for these positive outcomes, although only based on a limited number of available post audit studies, could be the increasing maturity of groundwater and hydrological modelling, e.g., the development of consistent modelling protocols and the focus on equifinallity issues. Another reason could be the change in logistical conditions, such as the increased computer power, enabling highly parameterized transient models, and inverse calibration methods that assess hundreds to thousands of model realizations.

Even in the formulation and pre-calibration part of the model protocol, the likelihood of catching model errors is much higher, because model testing has become more feasible in practice.

The original model predictions using the Baseline model were more or less consistent with the Post audit-Hydrogeology model. One explanation for this could be that the Baseline model was already based on a multi-objective calibration, and adding the drawdown and water volume objective did not change the optimization significantly. For instance, the calibration to annual groundwater fluctuation included in the Baseline model calibration would affect some of the same physical aquifer properties, e.g., specific yield, similarly to the calibration to the drawdown curves included in the Post audit-Hydrogeology model calibration.

Altogether, we consider post audits an important method for testing the performance of a model, documenting the model's robustness, and hence enhancing the credibility of model predictions.

### 4.3. Good Practice for Post Audits

Anderson and Woessner [4] inserted post audits into the general modelling protocol, to take place after validation of the model and after a period of time had passed over which the model predictions could be compared to the actual conditions. The post audit could then generate model knowledge that could feed into the conceptual model and therefore end in an improved iteration of most of the steps in the modelling protocol. A direct protocol for a post audit has, to the knowledge of the authors, not previously been proposed. A first attempt towards developing a protocol could be to adopt the four-step procedure that was used in this study. Regarding the terminology used in the protocol, a post audit can be performed on an original model, e.g., test original model predictions, or a post audit model can be developed, which is a new model developed from data produced in a period after the original model. The proposed model protocol involves four steps:

Step 1: Perform validation tests of predictions with the original model; e.g., in the Silkeborg case, test simulations with newly observed drawdown data. This step results in the original results being post audited.

Step 2: Estimate the predictive uncertainty of the original model, e.g., applying a Monte Carlo methodology.

Step 3: Update and revise the original model, resulting in the production of a post audit model. This step includes a possible revision of the conceptual model (e.g., in this case, a revision of the geology) and a revision of the calibration (e.g., a change of the objective functions based on new knowledge and re-calibration). This step produces a post audit model. In this study, two generations of post audit models were produced, one calibrated with the original objective function but with revised geology, and one calibrated with both a revised objective function and geology.

Step 4: Perform predictions and uncertainty assessments with the post audit model(s). Step 4 is a repeat of step 2, but for the post audit model(s).

Based on these steps, it can be evaluated whether the original model predictions should be repeated with a new post audit model, or if the original predictions were sufficiently robust.

The important new step in the definition of a modelling protocol is to include tests of the predictive uncertainty of the original and post audit models. This step is, not only of academic interest, but also relevant for management practice. For instance, in this modelling case, the original uncertainty was shown to be underestimated. In 2017, some of the permanent monitoring wells showed groundwater levels at the original modelled upper 95% confidence level. However, with the new understanding of a wider range of uncertainty, the model predictions were still trustworthy from a manager's point of view.

The literature on post audits of groundwater and hydrological model predictions is still very limited. Since the study by Anderson et al. [8] in 2018, only one peer-reviewed study on the post audit of groundwater model predictions has been published, by Flinders

et al. in 2022 [34]. Besides Karlsen et al. [17] and this study, we are not familiar with any studies assessing original uncertainty assessments in a post audit framework. In contrast to Karlsen et al. [17], this study showed an original underestimation of the predictive uncertainty. It would therefore be beneficial from an academic and practical perspective to have a full understanding of the uncertainty of model predictions. The proposed post audit protocol attempts to accomplish this.

**Author Contributions:** Conceptualization, J.K., J.C.R. and L.T.; methodology, J.K., J.C.R. and L.T.; software, LT.; validation, J.K. and L.T.; formal analysis, J.K.; investigation, J.K. and L.T.; resources, J.K., J.C.R. and L.T.; data curation, J.K. and L.T.; writing—original draft preparation, J.K.; writing—review and editing, J.K. and L.T.; visualization, J.K.; project administration, L.T.; funding acquisition, J.K., J.C.R. and L.T. All authors have read and agreed to the published version of the manuscript.

**Funding:** The study was funded by the Danish Road Directorate and the Geological Survey of Denmark and Greenland.

**Data Availability Statement:** The data presented in this study are available on request from the corresponding author.

**Acknowledgments:** The authors greatly appreciate the three reviewers' constructive comments and suggestions for improvements to the manuscript.

**Conflicts of Interest:** The authors declare no conflict of interest.

## Appendix A

**Table A1.** Calibration of the Baseline model.

| Parameter | No | Unit | Transformation | Initial Value | Estimated Value | 95 % Confidence Limits | |
|---|---|---|---|---|---|---|---|
| | | | | | | Lower | Upper |
| Horizontal hydraulic conductivity of glacial clay (till) | 1 | ms$^{-1}$ | log | $5.60 \times 10^{-6}$ | $1.80 \times 10^{-4}$ | $4.68 \times 10^{-17}$ | $6.91 \times 10^{8}$ |
| Vertical conductivity of glacial clay (till) | 2 | ms$^{-1}$ | tied #1 | $5.60 \times 10^{-8}$ | $1.80 \times 10^{-6}$ | | |
| Horizontal hydraulic conductivity of glacial sand | 3 | ms$^{-1}$ | log | $8.59 \times 10^{-5}$ | $2.21 \times 10^{-4}$ | $6.53 \times 10^{-5}$ | $7.50 \times 10^{-4}$ |
| Vertical conductivity of glacial sand | 4 | ms$^{-1}$ | tied #3 | $8.59 \times 10^{-6}$ | $2.21 \times 10^{-5}$ | | |
| Horizontal hydraulic conductivity of Miocene clay | 5 | ms$^{-1}$ | log | $4.57 \times 10^{-6}$ | $5.02 \times 10^{-6}$ | $7.65 \times 10^{-7}$ | $3.30 \times 10^{-5}$ |
| Vertical hydraulic conductivity of Miocene clay | 6 | ms$^{-1}$ | tied #5 | $4.57 \times 10^{-7}$ | $5.02 \times 10^{-7}$ | | |
| Horizontal hydraulic conductivity of Miocene sand (upper) | 7 | ms$^{-1}$ | log | $1.51 \times 10^{-3}$ | $7.02 \times 10^{-4}$ | $2.43 \times 10^{-4}$ | $2.03 \times 10^{-3}$ |
| Vertical conductivity of Miocene sand (upper) | 8 | ms$^{-1}$ | tied #7 | $1.51 \times 10^{-4}$ | $7.02 \times 10^{-5}$ | | |
| Horizontal hydraulic conductivity of Miocene sand (lower) | 9 | ms$^{-1}$ | log | $7.48 \times 10^{-5}$ | $4.75 \times 10^{-5}$ | $2.85 \times 10^{-5}$ | $7.91 \times 10^{-5}$ |
| Vertical conductivity of Miocene sand (lower) | 10 | ms$^{-1}$ | tied #9 | $7.48 \times 10^{-6}$ | $4.75 \times 10^{-6}$ | | |
| Horizontal hydraulic conductivity of terrace sand | 11 | ms$^{-1}$ | log | $1.50 \times 10^{-3}$ | $1.06 \times 10^{-3}$ | $6.25 \times 10^{-4}$ | $1.81 \times 10^{-3}$ |
| Vertical hydraulic conductivity of terrace sand | 12 | ms$^{-1}$ | log | $1.50 \times 10^{-4}$ | $1.76 \times 10^{-4}$ | $9.69 \times 10^{-48}$ | $3.19 \times 10^{39}$ |
| Horizontal conductivity of test p. areas of the terrace sand | 13 | ms$^{-1}$ | fixed | $1.50 \times 10^{-3}$ | | | |
| Vertical conductivity of test p. areas of the terrace sand | 14 | ms$^{-1}$ | fixed | $1.50 \times 10^{-4}$ | | | |
| Horizontal hydraulic conductivity of sand topsoil ˆ | 15 | ms$^{-1}$ | tied #11 | $1.25 \times 10^{-3}$ | $8.85 \times 10^{-4}$ | | |
| Vertical conductivity of sand topsoil ˆ | 16 | ms$^{-1}$ | tied #11 | $1.25 \times 10^{-4}$ | $8.85 \times 10^{-5}$ | | |
| Horizontal hydraulic conductivity of peat topsoil ˆ | 17 | ms$^{-1}$ | tied #1 | $7.07 \times 10^{-5}$ | $2.27 \times 10^{-3}$ | | |
| Vertical conductivity of peat topsoil ˆ | 18 | ms$^{-1}$ | tied #1 | $7.07 \times 10^{-6}$ | $2.27 \times 10^{-4}$ | | |
| Horizontal hydraulic conductivity of clay topsoil ˆ | 19 | ms$^{-1}$ | tied #1c | $5.60 \times 10^{-6}$ | $1.80 \times 10^{-4}$ | | |
| Vertical conductivity of clay topsoil ˆ | 20 | ms$^{-1}$ | tied #1 | $5.60 \times 10^{-8}$ | $1.80 \times 10^{-6}$ | | |
| Specific yield of glacial clay (till) | 21 | * | log | 0.20 | 0.19 | 0.10 | 0.39 |
| Specific yield of glacial sand | 22 | * | tied #21 | 0.20 | 0.19 | | |
| Specific yield of Miocene clay | 23 | * | tied #21 | 0.20 | 0.19 | | |
| Specific yield of Miocene sand (upper) | 24 | * | tied #21 | 0.20 | 0.19 | | |
| Specific yield of Miocene sand (lower) | 25 | * | tied #21 | 0.20 | 0.19 | | |
| Specific yield of terrace sand | 26 | * | log | 0.50 | 0.50 | 0.29 | 0.85 |
| Specific yield of test pumping areas | 27 | * | tied #26 | 0.50 | 0.50 | | |
| Specific yield of sand topsoil ˆ | 28 | * | tied #26 | 0.50 | 0.50 | | |
| Specific yield of clay topsoil ˆ | 29 | * | tied #21 | 0.20 | 0.19 | | |
| Specific yield of peat topsoil ˆ | 30 | * | tied #21 | 0.20 | 0.19 | | |
| Specific storage of glacial clay | 31 | m$^{-1}$ | fixed | $1.23 \times 10^{-5}$ | | | |
| Specific storage of glacial sand | 32 | m$^{-1}$ | fixed | $2.74 \times 10^{-5}$ | | | |
| Specific storage of Miocene clay | 33 | m$^{-1}$ | fixed | $1.23 \times 10^{-5}$ | | | |

**Table A1.** *Cont.*

| Parameter | No | Unit | Transformation | Initial Value | Estimated Value | 95 % Confidence Limits | |
|---|---|---|---|---|---|---|---|
| | | | | | | Lower | Upper |
| Specific storage of Miocene sand (upper) | 34 | $m^{-1}$ | fixed | $2.74 \times 10^{-5}$ | | | |
| Specific storage of Miocene sand (lower) | 35 | $m^{-1}$ | fixed | $2.74 \times 10^{-5}$ | | | |
| Specific storage of terrace sand | 36 | $m^{-1}$ | fixed | $2.74 \times 10^{-5}$ | | | |
| Specific storage of test pumping areas | 37 | $m^{-1}$ | fixed | $2.74 \times 10^{-5}$ | | | |
| Specific storage of sand topsoil ^ | 38 | $m^{-1}$ | fixed | $2.74 \times 10^{-5}$ | | | |
| Specific storage of peat topsoil ^ | 39 | $m^{-1}$ | fixed | $1.23 \times 10^{-5}$ | | | |
| Specific storage of clay topsoil ^ | 40 | $m^{-1}$ | fixed | $1.23 \times 10^{-5}$ | | | |
| Drain constant for MIKE SHE SZ drains | 41 | $s^{-1}$ | log | $2.00 \times 10^{-7}$ | $1.09 \times 10^{-7}$ | $7.34 \times 10 \text{ x}^{-20}$ | $1.61 \times 10^5$ |
| Detention storage | 42 | mm | log | 4.74 | 4.63 | 1.83 | 11.72 |
| Overland flow manning number | 43 | $m^{-1/3}s^{-1}$ | log | 4.00 | 12.08 | 1.63 | 89.65 |
| Conductance for dynamic head boundary condition N | 44 | $m^{-2}s^{-1}$ | fixed | $1.00 \times 10^{-8}$ | | | |
| Conductance for dynamic head boundary condition E | 45 | $m^{-2}s^{-1}$ | fixed | $5.00 \times 10^{-7}$ | | | |
| Conductance for dynamic head boundary condition W | 46 | $m^{-2}s^{-1}$ | fixed | $1.00 \times 10^{-8}$ | | | |

Note: * Dimensionless parameter, # tied to No. $\times$ parameter, ^ 3 m thick topsoil unit.

**Table A2.** Calibration of Post audit-Geology model.

| Parameter | No | Unit | Transformation | Initial Value | Estimated Value | 95 % Confidence Limits | |
|---|---|---|---|---|---|---|---|
| | | | | | | Lower | Upper |
| Horizontal hydraulic conductivity of glacial clay (till) | 1 | $ms^{-1}$ | log | $5.60 \times 10^{-6}$ | $9.00 \times 10^{-5}$ | $4.02 \times 10^{-22}$ | $2.02 \times 10^{13}$ |
| Vertical conductivity of glacial clay (till) | 2 | $ms^{-1}$ | tied #1 | $5.60 \times 10^{-8}$ | $9.00 \times 10^{-7}$ | | |
| Horizontal hydraulic conductivity of glacial sand | 3 | $ms^{-1}$ | log | $8.59 \times 10^{-5}$ | $1.77 \times 10^{-4}$ | $7.19 \times 10^{-5}$ | $4.33 \times 10^{-4}$ |
| Vertical conductivity of glacial sand | 4 | $ms^{-1}$ | tied #3 | $8.59 \times 10^{-6}$ | $1.77 \times 10^{-4}$ | | |
| Horizontal hydraulic conductivity of Miocene clay | 5 | $ms^{-1}$ | log | $4.57 \times 10^{-6}$ | $6.10 \times 10^{-7}$ | $3.78 \times 10^{-7}$ | $9.84 \times 10^{-7}$ |
| Vertical hydraulic conductivity of Miocene clay | 6 | $ms^{-1}$ | tied #5 | $4.57 \times 10^{-7}$ | $6.10 \times 10^{-7}$ | | |
| Horizontal hydraulic conductivity of Miocene sand (upper) | 7 | $ms^{-1}$ | log | $1.51 \times 10^{-3}$ | $1.64 \times 10^{-4}$ | $4.41 \times 10^{-5}$ | $6.12 \times 10^{-4}$ |
| Vertical conductivity of Miocene sand (upper) | 8 | $ms^{-1}$ | tied #7 | $1.51 \times 10^{-4}$ | $1.64 \times 10^{-4}$ | | |
| Horizontal hydraulic conductivity of Miocene sand (lower) | 9 | $ms^{-1}$ | log | $7.48 \times 10^{-5}$ | $6.85 \times 10^{-5}$ | $4.52 \times 10^{-5}$ | $1.04 \times 10^{-4}$ |
| Vertical conductivity of Miocene sand (lower) | 10 | $ms^{-1}$ | tied #9 | $7.48 \times 10^{-6}$ | $6.85 \times 10^{-5}$ | | |
| Horizontal hydraulic conductivity of terrace sand | 11 | $ms^{-1}$ | log | $1.50 \times 10^{-3}$ | $9.90 \times 10^{-4}$ | $6.95 \times 10^{-4}$ | $1.41 \times 10^{-3}$ |
| Vertical hydraulic conductivity of terrace sand | 12 | $ms^{-1}$ | log | $1.50 \times 10^{-4}$ | $2.98 \times 10^{-4}$ | $2.49 \times 10^{-27}$ | $3.57 \times 10^{19}$ |
| Horizontal conductivity of test p. areas of the terrace sand | 13 | $ms^{-1}$ | fixed | $1.50 \times 10^{-3}$ | | | |
| Vertical conductivity of test p. areas of the terrace sand | 14 | $ms^{-1}$ | fixed | $1.50 \times 10^{-4}$ | | | |

**Table A2.** *Cont.*

| Parameter | No | Unit | Transformation | Initial Value | Estimated Value | 95 % Confidence Limits Lower | Upper |
|---|---|---|---|---|---|---|---|
| Horizontal hydraulic conductivity of sand topsoil ˆ | 15 | ms$^{-1}$ | tied #11 | $1.25 \times 10^{-3}$ | $8.25 \times 10^{-4}$ | | |
| Vertical conductivity of sand topsoil ˆ | 16 | ms$^{-1}$ | tied #11 | $1.25 \times 10^{-4}$ | $8.25 \times 10^{-5}$ | | |
| Horizontal hydraulic conductivity of peat topsoil ˆ | 17 | ms$^{-1}$ | tied #1 | $7.07 \times 10^{-5}$ | $1.14 \times 10^{-3}$ | | |
| Vertical conductivity of peat topsoil ˆ | 18 | ms$^{-1}$ | tied #1 | $7.07 \times 10^{-6}$ | $1.14 \times 10^{-4}$ | | |
| Horizontal hydraulic conductivity of clay topsoil ˆ | 19 | ms$^{-1}$ | tied #1c | $5.60 \times 10^{-6}$ | $9.00 \times 10^{-5}$ | | |
| Vertical conductivity of clay topsoil ˆ | 20 | ms$^{-1}$ | tied #1 | $5.60 \times 10^{-8}$ | $9.00 \times 10^{-7}$ | | |
| Specific yield of glacial clay (till) | 21 | * | log | 0.20 | 0.12 | 0.08 | 0.17 |
| Specific yield of glacial sand | 22 | * | tied #21 | 0.20 | 0.12 | | |
| Specific yield of Miocene clay | 23 | * | tied #21 | 0.20 | 0.12 | | |
| Specific yield of Miocene sand (upper) | 24 | * | tied #21 | 0.20 | 0.12 | | |
| Specific yield of Miocene sand (lower) | 25 | * | tied #21 | 0.20 | 0.12 | | |
| Specific yield of terrace sand | 26 | * | log | 0.50 | 0.50 | 0.32 | 0.78 |
| Specific yield of test pumping areas | 27 | * | tied #26 | 0.50 | 0.50 | | |
| Specific yield of sand topsoil ˆ | 28 | * | tied #26 | 0.50 | 0.50 | | |
| Specific yield of clay topsoil ˆ | 29 | * | tied #21 | 0.20 | 0.12 | | |
| Specific yield of peat topsoil ˆ | 30 | * | tied #21 | 0.20 | 0.12 | | |
| Specific storage of glacial clay | 31 | m$^{-1}$ | fixed | $1.23 \times 10^{-5}$ | | | |
| Specific storage of glacial sand | 32 | m$^{-1}$ | fixed | $2.74 \times 10^{-5}$ | | | |
| Specific storage of Miocene clay | 33 | m$^{-1}$ | fixed | $1.23 \times 10^{-5}$ | | | |
| Specific storage of Miocene sand (upper) | 34 | m$^{-1}$ | fixed | $2.74 \times 10^{-5}$ | | | |
| Specific storage of Miocene sand (lower) | 35 | m$^{-1}$ | fixed | $2.74 \times 10^{-5}$ | | | |
| Specific storage of terrace sand | 36 | m$^{-1}$ | fixed | $2.74 \times 10^{-5}$ | | | |
| Specific storage of test pumping areas | 37 | m$^{-1}$ | fixed | $2.74 \times 10^{-5}$ | | | |
| Specific storage of sand topsoil ˆ | 38 | m$^{-1}$ | fixed | $2.74 \times 10^{-5}$ | | | |
| Specific storage of peat topsoil ˆ | 39 | m$^{-1}$ | fixed | $1.23 \times 10^{-5}$ | | | |
| Specific storage of clay topsoil ˆ | 40 | m$^{-1}$ | fixed | $1.23 \times 10^{-5}$ | | | |
| Drain constant for MIKE SHE SZ drains | 41 | s$^{-1}$ | log | $2.00 \times 10^{-7}$ | $9.51 \times 10^{-7}$ | $2.97 \times 10^{-8}$ | $3.04 \times 10^{-5}$ |
| Detention storage | 42 | mm | log | 4.74 | 4.22 | 1.68 | 10.59 |
| Overland flow manning number | 43 | m$^{-1/3}$s$^{-1}$ | log | 4.00 | 2.60 | 0.25 | 27.46 |
| Conductance for dynamic head boundary condition N | 44 | m$^{-2}$s$^{-1}$ | fixed | $1.00 \times 10^{-8}$ | | | |
| Conductance for dynamic head boundary condition E | 45 | m$^{-2}$s$^{-1}$ | fixed | $5.00 \times 10^{-7}$ | | | |
| Conductance for dynamic head boundary condition W | 46 | m$^{-2}$s$^{-1}$ | fixed | $1.00 \times 10^{-8}$ | | | |

Note: * Dimensionless parameter, # tied to No. x parameter, ˆ 3 m thick topsoil unit.

Table A3. Calibration of Post audit-Hydrogeology model.

| Parameter | No | Unit | Transformation | Initial Value | Estimated Value | 95 % Confidence Limits | |
|---|---|---|---|---|---|---|---|
| | | | | | | Lower | Upper |
| Horizontal hydraulic conductivity of glacial clay (till) | 1 | $ms^{-1}$ | log | $5.60 \times 10^{-6}$ | $3.26 \times 10^{-6}$ | $3.70 \times 10^{-9}$ | $2.87 \times 10^{-3}$ |
| Vertical conductivity of glacial clay (till) | 2 | $ms^{-1}$ | tied #1 | $5.60 \times 10^{-8}$ | $3.26 \times 10^{-8}$ | | |
| Horizontal hydraulic conductivity of glacial sand | 3 | $ms^{-1}$ | log | $8.59 \times 10^{-5}$ | $3.49 \times 10^{-5}$ | $4.66 \times 10^{-6}$ | $2.61 \times 10^{-4}$ |
| Vertical conductivity of glacial sand | 4 | $ms^{-1}$ | tied #3 | $8.59 \times 10^{-6}$ | $3.49 \times 10^{-6}$ | | |
| Horizontal hydraulic conductivity of Miocene clay | 5 | $ms^{-1}$ | log | $4.57 \times 10^{-6}$ | $3.89 \times 10^{-6}$ | $9.25 \times 10^{-7}$ | $1.64 \times 10^{-5}$ |
| Vertical hydraulic conductivity of Miocene clay | 6 | $ms^{-1}$ | tied #5 | $4.57 \times 10^{-7}$ | $3.89 \times 10^{-7}$ | | |
| Horizontal hydraulic conductivity of Miocene sand (upper) | 7 | $ms^{-1}$ | log | $1.51 \times 10^{-3}$ | $1.78 \times 10^{-3}$ | $7.12 \times 10^{-4}$ | $4.43 \times 10^{-3}$ |
| Vertical conductivity of Miocene sand (upper) | 8 | $ms^{-1}$ | tied #7 | $1.51 \times 10^{-4}$ | $1.78 \times 10^{-4}$ | | |
| Horizontal hydraulic conductivity of Miocene sand (lower) | 9 | $ms^{-1}$ | log | $7.48 \times 10^{-5}$ | $9.25 \times 10^{-5}$ | $5.56 \times 10^{-5}$ | $1.54 \times 10^{-4}$ |
| Vertical conductivity of Miocene sand (lower) | 10 | $ms^{-1}$ | tied #9 | $7.48 \times 10^{-6}$ | $9.25 \times 10^{-6}$ | | |
| Horizontal hydraulic conductivity of terrace sand | 11 | $ms^{-1}$ | log | $1.50 \times 10^{-3}$ | $1.64 \times 10^{-3}$ | $1.12 \times 10^{-3}$ | $2.39 \times 10^{-3}$ |
| Vertical hydraulic conductivity of terrace sand | 12 | $ms^{-1}$ | log | $1.50 \times 10^{-4}$ | $5.08 \times 10^{-5}$ | $1.16 \times 10^{-11}$ | $2.22 \times 10^{2}$ |
| Horizontal conductivity of test p. areas of the terrace sand | 13 | $ms^{-1}$ | fixed | $1.50 \times 10^{-3}$ | | | |
| Vertical conductivity of test p. areas of the terrace sand | 14 | $ms^{-1}$ | fixed | $1.50 \times 10^{-4}$ | | | |
| Horizontal hydraulic conductivity of sand topsoil ^ | 15 | $ms^{-1}$ | tied #11 | $1.25 \times 10^{-3}$ | $1.37 \times 10^{-3}$ | | |
| Vertical conductivity of sand topsoil ^ | 16 | $ms^{-1}$ | tied #11 | $1.25 \times 10^{-4}$ | $1.37 \times 10^{-4}$ | | |
| Horizontal hydraulic conductivity of peat topsoil ^ | 17 | $ms^{-1}$ | tied #1 | $7.07 \times 10^{-5}$ | $4.11 \times 10^{-5}$ | | |
| Vertical conductivity of peat topsoil ^ | 18 | $ms^{-1}$ | tied #1 | $7.07 \times 10^{-6}$ | $4.11 \times 10^{-6}$ | | |
| Horizontal hydraulic conductivity of clay topsoil ^ | 19 | $ms^{-1}$ | tied #1c | $5.60 \times 10^{-6}$ | $3.26 \times 10^{-6}$ | | |
| Vertical conductivity of clay topsoil ^ | 20 | $ms^{-1}$ | tied #1 | $5.60 \times 10^{-8}$ | $3.26 \times 10^{-8}$ | | |
| Specific yield of glacial clay (till) | 21 | * | log | 0.20 | 0.24 | 0.11 | 0.52 |
| Specific yield of glacial sand | 22 | * | tied #21 | 0.20 | 0.24 | | |
| Specific yield of Miocene clay | 23 | * | tied #21 | 0.20 | 0.24 | | |
| Specific yield of Miocene sand (upper) | 24 | * | tied #21 | 0.20 | 0.24 | | |
| Specific yield of Miocene sand (lower) | 25 | * | tied #21 | 0.20 | 0.24 | | |
| Specific yield of terrace sand | 26 | * | log | 0.50 | 0.50 | 0.39 | 0.64 |
| Specific yield of test pumping areas | 27 | * | tied #26 | 0.50 | 0.50 | | |
| Specific yield of sand topsoil ^ | 28 | * | tied #26 | 0.50 | 0.50 | | |
| Specific yield of clay topsoil ^ | 29 | * | tied #21 | 0.20 | 0.24 | | |
| Specific yield of peat topsoil ^ | 30 | * | tied #21 | 0.20 | 0.24 | | |
| Specific storage of glacial clay | 31 | $m^{-1}$ | fixed | $1.23 \times 10^{-5}$ | | | |
| Specific storage of glacial sand | 32 | $m^{-1}$ | fixed | $2.74 \times 10^{-5}$ | | | |
| Specific storage of Miocene clay | 33 | $m^{-1}$ | fixed | $1.23 \times 10^{-5}$ | | | |
| Specific storage of Miocene sand (upper) | 34 | $m^{-1}$ | fixed | $2.74 \times 10^{-5}$ | | | |

**Table A3.** *Cont.*

| Parameter | No | Unit | Transformation | Initial Value | Estimated Value | 95 % Confidence Limits Lower | Upper |
|---|---|---|---|---|---|---|---|
| Specific storage of Miocene sand (lower) | 35 | $m^{-1}$ | fixed | $2.74 \times 10^{-5}$ | | | |
| Specific storage of terrace sand | 36 | $m^{-1}$ | fixed | $2.74 \times 10^{-5}$ | | | |
| Specific storage of test pumping areas | 37 | $m^{-1}$ | fixed | $2.74 \times 10^{-5}$ | | | |
| Specific storage of sand topsoil ˆ | 38 | $m^{-1}$ | fixed | $2.74 \times 10^{-5}$ | | | |
| Specific storage of peat topsoil ˆ | 39 | $m^{-1}$ | fixed | $1.23 \times 10^{-5}$ | | | |
| Specific storage of clay topsoil ˆ | 40 | $m^{-1}$ | fixed | $1.23 \times 10^{-5}$ | | | |
| Drain constant for MIKE SHE SZ drains | 41 | $s^{-1}$ | log | $2.00 \times 10^{-7}$ | $1.18 \times 10^{-7}$ | $2.84 \times 10^{-9}$ | $4.89 \times 10^{-6}$ |
| Detention storage | 42 | mm | log | 4.74 | 7.62 | 2.80 | 20.71 |
| Overland flow manning number | 43 | $m^{-1/3}s^{-1}$ | log | 4.00 | 2.10 | 0.04 | 122.07 |
| Conductance for dynamic head boundary condition N | 44 | $m^{-2}s^{-1}$ | fixed | $1.00 \times 10^{-8}$ | | | |
| Conductance for dynamic head boundary condition E | 45 | $m^{-2}s^{-1}$ | fixed | $5.00 \times 10^{-7}$ | | | |
| Conductance for dynamic head boundary condition W | 46 | $m^{-2}s^{-1}$ | fixed | $1.00 \times 10^{-8}$ | | | |

Note: * Dimensionless parameter, # tied to No. x parameter, ˆ 3 m thick top soil unit.

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
