# Peer review of "Post Audit of Groundwater Model Predictions under Changing Conditions"

_water, doi:10.3390/w15061144_

Round 1
Reviewer 1 Report
This manuscript proposes a guideline to use post audits for groundwater modelling to ensure model validity for new time data sets.
The manuscript is well written and can be published following some minor modifications. Suggestions are:
L111. The period 2010-2019 can be 8,9 or 10 years. Specify.
L115. The timeline does not fit. The motorway should last 100 yr, it was finished in 2019 so it should last until 2119. Yet, the climate change is regarded only until 2100 (L116). Clarify.
L172. And other locations in the manuscript. It should be specified what the authors mean by "model". To me, the term "model" is often used without specification. Conceptual model? Flow model? Hydrostratigraphic model? Geological model?
L176ff. Follow-up on the previous comment: It would be nice to simply give each model a name and always use that name when refering to a specific model. In general, use of the word "model" has to be improved.
Section 2.5. Are layers=element layers? If so, then 3 element layers ("numerical layers") seems little, given that unsaturated flow is also simulated. Also, about the spatial discretization, have other spatial discretizations been tested? Have the authors conducted a grid convergence test? Are grid effects excluded?
Author Response
We thank the reviewer for all the constructive comments.
Line numbers refer to the revised manuscript (without track changes)
(L111): As suggested. We have specified the period as suggested to nine years in the manuscript. (now L109)
(L115-116): We agree with the reviewer and have changed the text. The motorway should last until the end of the century and not 100 years from 2019. (now L 113-114).
(L172 and 176ff): We have gone through all the mentionings of the word “model” to clarify the exact meaning at the different locations in the manuscript.
(Section 2.5) Yes, the 3 layers are what could be named element layers. Simulation of the unsaturated zone is although not a part of the simulation in the 3 so-called numerical layers. The 3 layers only cover the saturated zone of the hydrological model. Actually, tests have been done with different model grid sizes. For instance, the study from 2015 (Kidmose, J., L. Troldborg, J.C. Refsgaard, N. Bischoff. 2015. Coupling of a distributed hydrological model with an urban stormwater model for impact analysis of forced infiltration. Journal of hydrology 525, 506–520.) used a model grid size of 50 by 50 m. Also, the study from 2019 used a different grid size than the one used in this study because the former studies were not related to the simulation of the drawdown curves during the construction phase. Information on these tests has been updated to section 2.5.
We thank the reviewer for all the constructive comments.
Reviewer 2 Report
This article is a case study. The authors use MIKESHE to Modelling the hydraulic head and discharge before and after a motorway construction (post audit modelling). The writing is clear and organized. I have no question.
Author Response
Thanks for taking the time to review our manuscript, it is very much appreciated.
Reviewer 3 Report
Post Audit of Groundwater Models Predictions Under Changing Conditions
Good to read this paper. It was well written; however, there are slips.
1. Why do you have to repeat the same affiliation 3 times. You could have written once and numbered it as 1.
2. The novelty of the research work or research gap has to be presented in the abstract. Why did you want to do this research work?
3. Introduction is quite okay with good number of paper citations. However, only a couple of papers are recently published papers. Therefore, authors are missing what has been done along the same thread in the recent past. Thus, I want authors to work on new literature and refer them. Probably to add a new paragraph of what has been done recently.
4. " Altogether single groundwater head values from 97 wells and daily values for more than two years of groundwater heads from 35 wells and discharge from the four gauging stations were collected and used in the original hydrological model" - Very interesting. However, can you also state the time period which you have used for these data? From year to year?
5. What is the seasonal effect to the study? The authors said that they have used daily data. Therefore, there should be some differences with respect to the winter and so on. How did you work on those?
6. Remove this "E" term from paper and use 10x
7. Can you have a separate section for Conclusions? Now it is in the discussion and everything are kinda mixed.
Good work!
Author Response
Thanks for taking the time to review our paper, it is very much appreciated.
Line numbers refer to the revised manuscript (without track changes)
- The repetitions have been changed – good suggestion.
- The abstract has been extended/changed to include our research motivation.
- The reviewer points out one of our motivations for doing the study. There are simply only very few post audits of quantitative groundwater models and original model predictions to be found in the scientific literature. Based on the suggestion by the reviewer, we have re-analysed the available scientific literature (via web of science) to present the newest landmarks within post audit of GW models/hydrological models. This analysis showed that we only missed one new study by Flinders et al. published in 2022. Previous literature on the subject was already discussed in the manuscript (have we confirmed). We have updated the manus with a discussion of this in the discussion part (now lines 570-578).
- Done as suggested. We have updated the information of the period of 2010-2012 (now lines 151-154) into the mentioned sentence by the reviewer. This refers to the period where the daily time series of the hydraulic head was used for the original model calibration.
- Our model is continuously run, simulating every day with daily input of precipitation, temperature and reference evapotranspiration. Maybe it was not clear in the original manuscript and there we have updated including the following “With simulations of daily changes in groundwater recharge, groundwater levels and stream flow through several years, in the original model 3 years of dynamic (non-steady state) simulations from 2010-2012, and later until the end of 2016, the models represent seasonal changes as well.” (now lines 166-173)
- We have searched the manuscript for E and replaced it with *10 as suggested. Changes were only found in the appendix.
- In order not to repeat points made in the discussion in a separate conclusion section we have decided to combine these. We also acknowledge that point was not made by any of the other two reviewers and therefore we have kept a combined section.
Again, thanks a lot for taking the time to review our manuscript and help us improve it.